# Molecular Mechanisms of Lymph Node Metastasis in Gallbladder Cancer: Insights into the Tumor Microenvironment

**DOI:** 10.3390/biomedicines13061372

**Published:** 2025-06-04

**Authors:** Qingyu Tang, Yichen Guan, Yubo Ma, Qi Li, Zhimin Geng

**Affiliations:** Department of Hepatobiliary Surgery, The First Affiliated Hospital of Xi’an Jiaotong University, Xi’an 710061, China; tangqingyu@stu.xjtu.edu.cn (Q.T.); guanyich@163.com (Y.G.); zxxy93747@163.com (Y.M.)

**Keywords:** gallbladder cancer, lymph node metastasis, tumor microenvironment, non-coding RNAs, vascular endothelial growth factor

## Abstract

Gallbladder cancer (GBC) is a highly aggressive malignancy with a propensity for lymph node metastasis (LNM), which significantly worsens prognosis. This review explores the molecular mechanisms underlying LNM in GBC, focusing on the roles of vascular endothelial growth factors (VEGFs), chemokines, cancer-associated fibroblasts (CAFs), tumor-associated macrophages (TAMs), hypoxia-inducible factors (HIFs), and non-coding RNAs (ncRNAs) in shaping the tumor microenvironment (TME). Unique features of GBC, such as its bile-rich microenvironment and hypoxia-driven lymphangiogenesis, are highlighted. We discuss how these factors promote lymphangiogenesis, immune evasion, and extracellular matrix (ECM) remodeling, collectively facilitating LNM. Potential therapeutic targets, including VEGF-C/D pathways, matrix metalloproteinase (MMP) inhibitors, and immune-modulating therapies, are also reviewed. Future research integrating single-cell omics and patient-derived organoid models is essential for advancing precision medicine in GBC.

## 1. Introduction

Gallbladder cancer (GBC) is the most common biliary tract malignant tumor, and ranks 6th in incidence among gastrointestinal malignant tumors, with poor prognosis, high malignancy, strong invasiveness, and a propensity for lymph node metastasis (LNM) [1]. There were approximately 122,462 new cases of GBC worldwide, accounting for 0.6% of all new cancer cases, and 89,031 deaths, representing 0.9% of all cancer-related deaths in 2022, particularly prevalent in regions such as South Asia and South America [2]. LNM is a hallmark of advanced-stage GBC and a central determinant of long-term prognosis and therapeutic decision-making [3,4,5]. Understanding the mechanisms underlying LNM in GBC is therefore of significant importance for improving patient outcomes.

Most GBC patients are diagnosed at an advanced stage with poor prognosis and high risk of LNM spread. It is early and widely known that the lymphatic drainage of GBC mainly passes through three pathways [6]. The right-sided pathway involves lymph flow along the common bile duct, reaching the LNs around the portal vein, and eventually draining into the para-aortic LNs [7]. This is the most common route observed in patients with LNM. The left-sided pathway passes through the LNs behind the pancreatic head, then moves to the hepatoduodenal ligament, and finally reaches the para-aortic LNs. The portal vein pathway allows lymph to drain directly through the hilar LNs into the para-aortic LNs [8]. Studies have shown that LNM is an independent factor affecting long-term survival in GBC, and proper LN dissection can improve patient survival [9].

The tumor microenvironment (TME) plays a crucial role in the initiation, progression, and metastasis of various cancers, including GBC [10]. The TME is a heterogeneous and dynamic ecosystem comprising tumor cells, immune cells, fibroblasts, endothelial cells, extracellular matrix (ECM) components, and signaling molecules (Figure 1) [10]. Unlike other gastrointestinal cancers, GBC LNM is uniquely influenced by the bile microenvironment, which alters immune response and promotes lymphatic dissemination. For instance, increased complement proteins in bile can drive tumor angiogenesis, and elevated bile acids induce cancer cell invasion via STAT3 signaling [11]. Chronic inflammation due to gallstones further creates an immunosuppressive niche that facilitates metastasis [11]. Emerging evidence suggests that specific molecules within the TME of GBC may drive LNM by specifically modulating key processes (Figure 2).

In this review, we discuss the molecules playing crucial roles in the LNM of GBC, also regarded as important components of TME. We discuss key points of the underlying mechanisms of how the important signaling pathways lead to LNM in GBC in order to provide new insights for developing targeted therapies and precisely applying neoadjuvant therapies.

## 2. Molecular Mechanisms Driving Lymph Node Metastasis in GBC

Building upon the complex interplay between tumor cells and TME, both cellular and soluble factors significantly contribute to the metastatic potential of GBC. Among the soluble factors, the specific roles of vascular endothelial growth factors (VEGFs) and chemokines as key regulators of cancer dissemination and LNM of GBC are widely reported.

### 2.1. Angiogenesis and Lymphangiogenesis

VEGFs are a family of endothelial growth factors that critically drive tumor angiogenesis and metastatic spread, including VEGF-A, VEGF-B, VEGF-C, VEGF-D, VEGF-E, and placental growth factors [12]. VEGF-A is essential for new blood vessel formation, while VEGF-B primarily supports vascular stability and facilitates fatty acid uptake in endothelial cells [13]. VEGF-C and VEGF-D are potent lymphangiogenic factors by interacting with receptor tyrosine kinases vascular endothelial growth factor receptor-2 (VEGFR-2) and vascular endothelial growth factor receptor-3 (VEGFR-3) on cell surfaces, initiating lymphangiogenesis, the formation of new lymphatic vessels from pre-existing structures [14].

Elevated levels of VEGF-C, VEGF-D, and their receptors, VEGFR-2 and VEGFR-3, are notably present at the invasive edge of GBC, correlating with increased LNM [15]. The GBC TME further modulates VEGF-C expression through tumor necrosis factor-alpha (TNF-α), which activates NF-κB-dependent VEGF-C signaling, thereby promoting lymphangiogenesis and GBC cell invasiveness [16]. Studies also indicate that cellular inhibitors of apoptosis protein 2 enhance VEGF-C expression via TNF-α-induced NF-κB signaling, thereby promoting GBC metastasis [17]. Additionally, the gallbladder’s bile-rich environment may intensify VEGF-driven lymphangiogenesis. Inflammatory mediators and bile acids in GBC can augment angiogenic signaling and lymphatic vessel formation, linking gallbladder physiology to increased metastatic spread [11].

VEGF-D, while less studied than VEGF-C, has emerged as a critical factor in GBC progression. Functional studies show that VEGF-D silencing in NOZ xenograft models reduces tumor proliferation, ascites, hepatic invasion, and lymphangiogenesis, resulting in complete inhibition of lymphatic metastasis [18]. These findings not only confirm VEGF-D’s role in remodeling the lymphatic vasculature but also support its direct effect on tumor cell behavior.

The dual mechanisms of VEGF-C and VEGF-D promote lymphangiogenesis and directly enhance invasiveness, positioning them as key factors in GBC metastasis. The overlapping roles of VEGF-C and VEGF-D in lymphangiogenesis suggest that simultaneous inhibition could provide enhanced therapeutic benefits, particularly given the TME’s dependence on VEGF-C/D signaling in GBC.

### 2.2. Chemokines and Their Receptors

Chemokine ligands and receptors play essential roles in directing various cell migration processes, including lymphatic homing and tumor cell dissemination. Anja Muller’s foundational work in 2001 elucidated the role of chemokines in guiding tumor cells toward LNs and lymphatic vessels, sparking extensive research into chemokine receptor functions in cancer metastasis [19].

C-X-C chemokine receptor 2 (CXCR2), particularly the C-X-C motif chemokine ligand 5 (CXCL5)/CXCR2 axis, is also crucial in tumor lymphatic metastasis and chemoresistance [20]. The CXCL5/CXCR2 axis promotes lymphangiogenesis, epithelial–mesenchymal transition (EMT), and matrix metalloproteinase (MMP) upregulation in cholangiocarcinoma (CCA) [21]. Beyond direct tumor cell signaling, CXCL5 secreted by fibroblasts, macrophages, Schwann cells, and mesenchymal stem cells can recruit CXCR2^+^ neutrophils, thereby enhancing immune evasion and metastatic niche formation [21]. While CXCR4 and CXCR2 are well studied in other cancers, their roles in GBC LNM remain underexplored, necessitating the use of functional in vivo models to delineate their contributions. Expanding on chemokine-mediated pathways, C-C chemokine receptor type 2 (CCR2) expression in GBC is associated with LNM and poor prognosis. Recently, Chen’s team found C-C motif chemokine ligand 2 (CCL2) bonded to CCR2 on tumor cells and activated the mitogen-activated protein kinase (MEK)–extracellular signal-regulated kinase (ERK)–ELK1 axis, which was found to correlate with LNM, leading to increased EMT induction and stemness of GBC in murine models [22].

In summary, chemokine receptors are crucial in GBC LN dissemination. These receptors promote tumor migration, immune evasion, stemness, and resistance to therapies such as gemcitabine. The heterogeneity of chemokine expression suggests that personalized therapeutic strategies may be necessary.

### 2.3. Hypoxia and Related Pathways

Building on the potential of targeting CAFs and M2 macrophages to inhibit immune evasion and metastasis, the role of physical properties within the TME warrants attention. Among these, hypoxia emerges as a crucial factor driving tumor progression and LNM in GBC [23]. Hypoxia, referring to reduced oxygen levels, commonly occurs in pathological states such as stroke, ischemia, inflammation, and the growth of solid tumors. The understanding of hypoxia’s influence on tumor biology began with Otto Warburg’s early 20th-century observations that cancer cells prefer glycolysis over oxidative phosphorylation [24]. Under hypoxic conditions, cancer cells activate hypoxia-inducible factors (HIFs), triggering a cascade of changes that drive tumor metastasis [25].

However, the role of HIF-1α in LNM in GBC remains controversial. Wu et al. [26] explored the relationship between the expression of HIF-1α and other related proteins with clinical parameters, including LN status in GBC tissues, and found no correlations. While HIF-1α is expressed in both GBC and normal gallbladder tissues, the expression of HIF-1α does not exhibit a direct, statistically significant correlation with LNM in GBC patients. These results suggest the irrelevance between HIF-1α and LNM of GBC.

By contrast, other studies have provided evidence supporting a pro-metastatic role of HIF-1α in GBC. For instance, knockdown of HIF-1α significantly reverses hypoxia-induced migration and downregulates the expression level of VEGFs in GBC, while the overexpression of HIF-1α is associated with LNM, suggesting HIF-1α may contribute to tumor migration via the HIF-1α/VEGF signaling pathway in GBC [27,28]. These findings suggest that HIF-1α may facilitate metastatic spread under certain conditions, possibly by promoting EMT or remodeling the lymphatic microenvironment.

Taken together, the discrepancy between studies demonstrates the complexity of hypoxia signaling in GBC. We hypothesize that hypoxia may contribute to LNM in GBC through at least two major pathways: (1) promoting EMT and enhancing migratory potential of tumor cells, and (2) remodeling the ECM and lymphangiogenesis via the HIF-1α/VEGF signaling axis. Clarifying these mechanisms will be crucial for determining whether hypoxia-related pathways represent viable therapeutic targets in GBC. Supporting this broader relevance, insights from other biliary tract cancers support the relevance of hypoxia-induced enzymes. For instance, hypoxia-induced procollagen-lysine, 2-oxoglutarate 5-dioxygenase 2 (PLOD2) impairs chemoresistance through EMT and is associated with poor prognosis and increased LNM, which is a significant unfavorable prognostic factor for predicting recurrence-free survival and OS in biliary tract cancer [29]. These findings further suggest that hypoxia may play a broader role across the biliary cancer spectrum, though tumor-specific differences remain to be elucidated.

### 2.4. Cell Adhesion and Migration Signaling

The ECM is a critical component of the TME, providing structural support while actively regulating tumor progression [30]. Its remodeling is mediated by enzymes such as MMPs and adhesion molecules like integrins.

#### 2.4.1. Matrix Metalloproteinases

MMPs are zinc-dependent endopeptidases that function as important ECM degradation components, affecting various cancer malignant processes, including LNM [31]. MMP-9 and MMP-2 are both gelatinases and significantly correlated with LNM in hepatocellular carcinoma [32], cervical cancer [33], breast cancer [34], and gastric cancer [35]. MMP-9 was first recognized as a prognostic factor related to LNM in intrahepatic cholangiocarcinoma (ICC) in 1999, with a higher expression indicating a higher risk of recurrence [36,37]. Horiuchi, H. (2004) claimed that the downregulation of MMP-2 and MMP-9 could prolong the survival of the mice with NOZ tumor by inhibiting LNM via impaired matrix-degrading ability [38]. Recent evidence suggests that MMP-2 and MMP-9 promote LNM in biliary tract cancers by facilitating EMT, enhancing ECM remodeling, and promoting lymphangiogenesis through paracrine interactions with stromal and lymphatic endothelial cells mediated via key signaling pathways including PDGF-p38/MAPK and calcium-sensing receptor-ERK axes [39,40,41]. More efforts are needed to understand the role of MMPs, especially gelatinases MMP-2 and MMP-9, in LNM, which may provide more insights into targeted therapies of GBC.

#### 2.4.2. Integrins

Integrins are adhesion molecule receptors have been widely reported to interact with VEGFs and other TME components, contributing to LNM of various cancers [42,43]. In humans, integrins comprise 8 β subunits and 18 α subunits, which together form 24 distinct integrin heterodimers [44]. Among them, the β6 subunit has been reported to be associated with ECM remodeling, cell invasiveness, and, therefore, metastasis in gastrointestinal cancers [45]. In CCA, integrin β6 has been reported to correlate with LNM and distant metastasis, increasing the secretion of MMP-9 [46]. Although similar evidence is currently lacking in GBC, this finding raises the possibility that integrin–MMP signaling may also contribute to lymphatic dissemination in GBC, which needs further investigation. The disruption of cell–cell and cell–matrix adhesion also plays a central role in GBC dissemination. CEACAM6, a cell adhesion molecule upregulated in GBC, promotes migration and invasion by interacting with Integrin β1 and PKCδ, triggering downstream activation of ERK and protein kinase B(AKT) pathways [47]. These signaling events reduce cellular adhesion and enhance motility, which may drive early steps of LNM.

#### 2.4.3. Fibronectins

Fibronectins (FNs) are ECM glycoproteins that bind to integrins and other matrix proteins, existing as a disulfide-linked dimer of nearly identical monomers [48]. FNs are found to be an important prognostic factor of LNM and related to an invasive phenotype of endometrioid endometrial cancer patients [49]. However, FNs are found to be irrelevant to LNM but associated with tumor cell proliferation and invasiveness in an Akt/mTOR/4E-BP1-signaling-pathway-dependent manner in GBC [50]. Specifically, FN treatment increases phosphorylation of FAK, Akt, mTOR, and downstream effector 4E-BP1, leading to enhanced protein synthesis, cell cycle progression, and invasion [50]. In addition to full-length fibronectin, in hepatocellular carcinoma, proteolytic fragments containing RGD and synergy motifs can enhance integrin α5β1-mediated signaling and contribute to tumor cell invasiveness [51,52]. These findings show FN’s multifaceted role in modulating tumor progression, both through full-length and proteolytically cleaved forms, and highlight the relevance of FN-associated signaling in various cancers [53]. More efforts are needed to understand the role of FNs in GBC metastasis process.

Taken together, ECM plays a significant role in cancer cell invasion, migration, and LNM. Inhibiting MMPs or blocking integrin–ECM interactions represent potential strategies for disrupting the metastatic cascade in GBC.

#### 2.4.4. Exosomes

Exosomes, membrane particles from 30 to 150 nm, carrying a range of bioactive molecules, actively participate in establishing pre-metastatic niches that facilitate lymphatic spread [54]. Hood et al. proposed that tumor-derived exosomes migrate to sentinel LNs, promoting lymphangiogenesis and creating a favorable environment for lymphatic invasion [55].

Furthermore, exosomes play a critical role in remodeling the ECM, primarily through MMPs and integrins, supporting the invasion of cancer cells into lymphatic vessels. For instance, exosomes enriched with integrin α6 interact with lymphatic endothelial cells via the integrin α6/CD151 axis in bladder cancer (BCa), inducing lymphatic remodeling and enhancing tumor cell adhesion and colonization in LNs [56]. While these mechanisms have been primarily characterized in cancers such as BCa, they may also operate in GBC. However, their functional relevance in GBC remains to be confirmed by future studies.

In addition, exosome-associated long non-coding RNAs (lncRNAs) play a notable role in lymphatic metastasis. Specific lncRNAs, such as LNMAT2 [57], TTN-AS1 [54], and TRPM2-AS [58], have been associated with LNM and increased vascular density across multiple cancers, implicating the significant role of exosomal lncRNAs in promoting metastatic spread.

These findings collectively underscore the role of exosomes in modulating the TME and facilitating lymphatic metastasis through diverse mechanisms. Targeting exosome-mediated pathways presents a promising therapeutic avenue for inhibiting lymphatic metastasis and improving patient outcomes in GBC.

### 2.5. Non-Coding RNAs

Non-coding RNAs (ncRNAs), including lncRNAs, microRNAs (miRNAs), and circular RNAs (circRNAs), have emerged as critical mediators in cancer progression and metastasis, often functioning as cargo within exosomes [59]. By bridging intercellular communication, ncRNAs regulate diverse molecular mechanisms that underlie GBC metastatic dissemination (Table 1).

#### 2.5.1. Long Non-Coding RNAs

LncRNAs, a class of ncRNAs longer than 200 nucleotides, have been increasingly recognized for their roles in cancer progression [71]. In GBC, several lncRNAs have been identified as key regulators of tumor growth, invasion, LNM, and prognosis through various molecular mechanisms. A systematic review identifies 15 lncRNAs related to GBC, among which LINC00152, HEGBC, MALAT1, and ROR are positively correlated with LNM, while GCASPC, MEG3, LET, and UCA1 show negative correlations [63]. Additionally, lncRNA H19 is positively correlated with LNM and unfavorable clinical outcomes in GBC patients, promoting EMT [60] and invasiveness of GBC cells mainly via the interaction with miR-342-3p and the Wnt/β-catenin pathway [61]. Recently, lncRNA LINC00662 has been found to be associated with LNM and tumor size in GBC, promoting EMT by LINC00662/miR-335-5p/OCT4 axis [62]. These insights suggest that lncRNAs are not only valuable as potential biomarkers for prognosis but also as therapeutic targets, particularly for interfering with key pathways like EMT, miRNA sponging, and chemoresistance. Further research should focus on elucidating the mechanisms driving lncRNA dysregulation in GBC and developing strategies for clinical benefit.

#### 2.5.2. MicroRNAs

MiRNAs have emerged as critical regulators in the progression and LNM of GBC, offering promising opportunities for diagnosis, prognosis, and therapy. These short ncRNAs influence gene expression and play significant roles in cancer cell proliferation, apoptosis, migration, invasion, and metastasis. Many miRNAs act as tumor suppressors and show a negative correlation with LNM in GBC. For example, miR-324-5p is significantly downregulated in GBC tissues and cells and acts as a tumor suppressor by inhibiting cell migration, invasion, and EMT by targeting TGF-β2 [64]. However, plasma miR-141 is elevated and positively correlated with LNM in GBC patients, promoting the proliferation and inhibiting the apoptosis of GBC cells [65]. These findings highlight the dual roles of miRNAs as both tumor suppressors and oncogenes in GBC. A deeper understanding of the miRNA regulatory network could not only reveal the molecular drivers of LNM but also provide a foundation for the development of miRNA-based biomarkers and therapeutic strategies.

#### 2.5.3. Circular RNAs

As the RNA sequencing technology rapidly advances, more and more circRNAs are identified. circRNAs are the products of pre-mRNA back-splicing of exons and take part in various biological processes [72]. CircPVT1 is highly expressed in GBC and correlates with LNM, acting as a sponge for miR-339-3p to upregulate MCL1 and promote GBC invasiveness [66]. Likewise, circTP63 correlates with LNM and poor prognosis, promoting EMT and invasion via miR-217 sequestration [67]. Beyond these, recent studies have uncovered additional oncogenic circRNAs in GBC. For example, circAATF is upregulated in GBC and elevates programmed cell death ligand 1 (PD-L1) expression by sponging miR-142-5p, thereby promoting immune evasion and enhancing responsiveness to anti-PD-L1 therapy [68]. CircEZH2, frequently overexpressed in GBC, drives tumor progression by functioning as a ceRNA for miR-556-5p, thereby upregulating SCD1 expression [69]. This axis facilitates the conversion of saturated fatty acids to monounsaturated fatty acids, reprogramming lipid metabolism to enhance membrane fluidity, suppress lipid peroxidation, and inhibit ferroptosis [69]. Similarly, circRNA NGFR has been shown to suppress ferroptotic cell death by upregulating GPX4 and maintaining mitochondrial function; its inhibition leads to Fe^2+^ accumulation, increased ROS levels, HO-1/FTH pathway activation, and ultimately ferroptotic cell death [70]. Taken together, these findings highlight that circRNAs regulate GBC progression through complex control of lipid metabolism and ferroptosis. However, significant gaps remain in our understanding, particularly regarding how these circRNAs are selectively expressed in GBC, what triggers their upregulation, and whether they interact with other RNA regulatory networks. Future efforts should focus on uncovering additional clinically relevant circRNAs and designing circRNA-targeting strategies that can inhibit their oncogenic roles in GBC progression and LNM.

In summary, ncRNAs and the interplay between these ncRNAs add complexity to the LNM process in GBC, as seen in the dual roles of tumor-suppressive and oncogenic ncRNAs. While the potential of ncRNAs as biomarkers and therapeutic targets is promising, significant challenges remain in fully elucidating their functional crosstalk and dysregulation in the TME. We believe that future research should adopt an integrative approach combining high-throughput sequencing and advanced functional validation to explore the therapeutic potential of ncRNAs.

### 2.6. Non-Canonical Regulators: Ubiquitin and Epigenetic Modulators

In addition to classical pathways, recent studies have highlighted the roles of novel regulatory proteins in GBC. TRIM47, a member of the tripartite motif-containing E3 ligase family, has been shown to promote tumor cell proliferation and migration by stabilizing PARP1 through K63-linked ubiquitination, thereby enhancing AKT pathway activity. Elevated TRIM47 levels are associated with poor prognosis and LNM, suggesting its role in driving tumor aggressiveness via post-translational modification [73]. Similarly, BRD9, an epigenetic reader involved in chromatin remodeling, was found to be upregulated in GBC tissues. It enhances tumor cell growth by partnering with the transcription factor FOXP1 to activate the PI3K-AKT signaling cascade. Inhibition of BRD9 led to reduced tumor growth in preclinical models, highlighting its therapeutic potential [74]. Collectively, these findings highlight a network of non-canonical regulators that converge on survival, migration, and EMT pathways, thereby influencing lymphatic spread, and may represent promising targets for intervention beyond classical signaling factors.

### 2.7. Metabolic Regulation

Metabolic remodeling plays a critical role in GBC metastasis, particularly through dysregulated glucose and lipid metabolism, which enhances tumor invasiveness and promotes immune evasion. Recent studies have shown that PP4R1 enhances glycolytic flux by activating the ERK1/2-PKM2 signaling axis, thereby promoting GBC cell proliferation and metastatic potential [75]. In parallel, LncBCL2L11 augments fatty acid oxidation by disrupting THOC5, which leads to the accumulation of acylcarnitines and subsequent stabilization of the lncRNA via m6A methylation—a feedforward loop facilitating GBC progression [76].

Lipid metabolism also plays a crucial role in shaping the GBC immune microenvironment. Fatty-acid-binding protein 1 (FABP1) is upregulated in liver-invasive regions of GBC and strongly correlates with LNM and poor prognosis. Notably, elevated FABP1 expression is inversely associated with CD8^+^ T cell infiltration, indicating a link between lipid metabolism and immune suppression [77]. This concept is further substantiated by studies in CCA, where a positive feedback loop involving oleic acid, PPARγ, and FABP4 promotes fatty acid uptake and oxidation, facilitating lymph node colonization and establishing an immunosuppressive niche via kynurenine production [78]. Moreover, elevated expression of fatty acid synthase (FASN), a key enzyme in de novo lipogenesis, contributes to GBC cell proliferation, migration, and chemoresistance by activating the PI3K/AKT signaling pathway [79]. This is supported by broader oncologic evidence showing that FASN overexpression drives PI3K/mTOR signaling and immune evasion in hepatic cell lines, and that inhibition of FASN leads to the degradation of downstream effectors through proteasomal and mTOR-dependent mechanisms [80].

Collectively, these findings highlight the critical role of glucose and lipid metabolic reprogramming in promoting GBC metastasis and immune evasion, and suggest that targeting key metabolic enzymes or pathways may offer therapeutic benefits in restraining nodal dissemination.

## 3. Tumor Microenvironment and Immune Regulation

### 3.1. Cancer-Associated Fibroblasts

Cancer-associated fibroblasts (CAFs), a resilient stromal cell subtype that contributes to tumor recurrence, have been extensively implicated in facilitating LNM primarily by secreting cytokines and inducing ECM remodeling across various cancer types [81,82].

In GBC, CAF-derived thrombospondin-4 binds to integrin α2 on GBC cells, activating the Akt-mediated heat shock factor 1 (HSF1) signaling pathway. This activation facilitates not only GBC cell proliferation but also EMT and the acquisition of cancer-stem-cell-like properties [83]. Meanwhile, the activation of HSF1 promotes the secretion of transforming growth factor-beta (TGF-β), which induces the redifferentiation of normal fibroblasts into CAFs. This feedback loop presents potential therapeutic targets for disrupting tumor progression and metastasis. In CCA, the work of Yan et al. highlights that CAF-mediated lymphangiogenesis promotes LNM via the secretion of platelet-derived growth factor-BB, further highlighting CAFs’ importance in cancer progression [84]. In ICC, alpha-smooth muscle actin and heparan sulfate 6-O-sulfotransferase 1 expression in CAFs have been linked to LNM and poor prognosis [85].

Notably, CAFs also contribute to lymphangiogenesis and metastasis in BCa, demonstrating the broad relevance of CAF-targeting strategies. In BCa, a specific CAF subset expressing PDGFRα and integrin alpha 11 (ITGA11) has been shown to interact with lymphatic endothelial cells through the interaction between ITGA11 on the CAFs and its receptor E-selectin on lymphatic endothelial cells [86]. Meanwhile, CAFs contribute to ECM remodeling by activating MMP2 to assist cancer cell infiltration into lymphatic vessels, thereby enhancing lymphovascular invasion and LNM. This study utilizes advanced methodologies to highlight the therapeutic potential of identifying and targeting specific CAF clusters to inhibit early-stage cancer metastasis [86].

Given these insights, therapeutic strategies aimed at disrupting the bidirectional CAF-tumor feedback loop, such as inhibiting TGF-β signaling, hold promise for mitigating LNM in GBC. The complexity of CAF heterogeneity within the TME needs further investigation into CAF-driven processes for more precise interventions.

### 3.2. Macrophages

Macrophages are key components of the tumor immune landscape, which are predominantly infiltrated by M2-type rather than M1-type macrophages [87]. M2 macrophages facilitate cancer cell metastasis, especially LNM, through several mechanisms, including secreting VEGFs, MMPs, and CCL2 [88,89,90]. In GBC, the bile-rich microenvironment exacerbates M2 macrophage recruitment and activation, enhancing its immunosuppressive effects through chemokine secretion [91]. M2 macrophages secrete CCL2, which binds to CCR2 on tumor cells and activates the MEK–ERK–ELK1 axis, leading to increased SNAIL expression and a more invasive phenotype of GBC [22].

Moreover, GBC stem-like cells can, in turn, promote the recruitment and polarization of macrophages toward the M2-like phenotype, forming a pro-metastatic feedback loop. On the other hand, M2 macrophages contribute to immune evasion through the secretion of chitinase-3-like protein 1 (YKL-40) [92], a chitinase-like glycoprotein, widely reported to promote tumor angiogenesis, immune suppression, and malignancy across various cancers and associated with increased PD-L1 expression on tumor cells [93,94]. Specifically, YKL-40 expression by M2-like macrophages has been found to correlate with LNM and GBC progression, partly through immune evasion mechanisms mediated by upregulated PD-L1 expression [91]. From another perspective, the knockdown of apoptosis-stimulating protein of p53-2 enhances M2 macrophage recruitment and promotes cancer metastasis through the atypical protein kinase C iota (aPKC-ι)/glioma-associated homologue-1 (GLI-1) signaling pathway, accompanied by increased secretion of chemokines and TNF-α, ultimately driving LNM [95]. Additionally, M2 macrophages release immunosuppressive cytokines like IL-10 and TGF-β, further inhibiting immune recognition and supporting tumor survival and dissemination [96]. These insights emphasize the importance of immune–tumor crosstalk triggered by M2 macrophages in facilitating LNM and offer potential therapeutic opportunities. Therapeutic strategies should focus on selective reprogramming of M2 cells toward a tumor-suppressive phenotype. For example, blocking key immunosuppressive factors such as YKL-40 or CCL2, or modulating metabolic pathways that maintain M2 polarization, could mitigate LNM while minimizing systemic inflammation [91].

### 3.3. Dendritic Cells

Recent investigations have begun to clarify the role of dendritic cells (DCs) in GBC, particularly in the context of LNM [97,98]. A study focusing on CD1a^+^ monocyte-derived DCs (CD1a-DCs) demonstrated that their infiltration into regional lymph nodes was significantly associated with LNM in GBC, with all patients in the CD1a-DCs-high group exhibiting nodal metastasis [99]. Notably, the presence of CD1a^+^ DCs in primary tumors, rather than metastatic LNs, showed a more pronounced correlation with surgical outcomes, implying that tumor-derived monocytes may give rise to DCs that promote nodal colonization [99].

These findings complement the established roles of TAMs and CAFs, expanding the spectrum of immune-modulatory cells involved in GBC progression. Moreover, they suggest the site-specific behavior of DCs in GBC versus other solid tumors, advocating for further investigation into how tumor histology influences immune regulation.

## 4. Discussion

Most GBC patients are diagnosed at an advanced stage with poor prognosis and high risk of LNM spread. On the microscopic scale, there is a lack of systematic studies of LNM in GBC at the molecular level. Understanding the mechanisms underlying LNM in GBC is crucial for improving patient outcomes and guiding treatment strategies. This review discusses not only the roles of various factors, including VEGFs and chemokines, HIFs, ECM components, and tumor immune-related cells, in promoting metastasis through interactions within the TME, but also the bidirectional crosstalk between tumor cells and TME components, such as CAF-driven ECM remodeling and TAM-mediated immune suppression. These findings highlight the dynamic interactions within the TME, which contribute to lymphangiogenesis and the formation of a pre-metastatic niche, and finally turn into LNM. Although not addressed in a standalone section, EMT is a recurring theme throughout the mechanisms discussed in this review. Multiple drivers of LNM in GBC—including chemokines, ncRNAs, CAFs, and macrophages—have been shown to promote EMT, thereby enhancing cancer cell migration and invasion. These molecular events converge to facilitate dissemination into lymphatic vessels.

Despite advancements in understanding these mechanisms, several knowledge gaps remain. For instance, the specific contributions of chemokine receptors, other tumor immune-related cells, like myeloid-derived suppressor cells, circulating tumor cells and regulatory T cells, in LNM have been well studied in other cancers, such as Bca [56], but their specific relevance in GBC remains poorly characterized. Similarly, physical factors in the TME, such as mechanical stress and pH levels, have been extensively explored in other cancer types. However, there is still a lack of high-impact studies and significant findings on these factors in GBC. Moreover, mutations in tumor suppressor genes such as TP53 and oncogenes like KRAS may interact with hypoxia signaling or ncRNA networks to modulate the TME and metastatic potential. However, their direct contribution to lymphatic dissemination in GBC awaits further investigation. Future studies are needed to clarify whether these cell populations and genetic mutations significantly contribute to nodal dissemination in GBC.

On the translational scale, the integration of advanced methodologies such as single-cell RNA sequencing [100], spatial transcriptomics [101], and patient-derived organoid models offers unprecedented opportunities to explore the complexity of LNM in GBC [102]. Recent single-cell RNA sequencing studies, for instance, have revealed heterogeneous immunosuppressive cell populations in GBC and identified novel drivers like midkine that foster pro-metastatic macrophage differentiation [100]. Spatial transcriptomics combined with organoid models can further pinpoint metastasis-associated genes. For example, repression of SEMA4A was shown to enhance GBC cell migration and survival [101]. Integrative multi-omics analyses of patient-derived GBC organoids are also being leveraged to uncover actionable vulnerabilities for metastasis [102]. These approaches could reveal novel therapeutic targets and biomarkers, thereby facilitating the development of precision medicine strategies for GBC patients. Novel machine learning models have also shown that metastatic lymph nodes are significantly associated with the prediction of survival, stratifying risk, and early recurrence of GBC patients [5,103]. These approaches not only refine prognostic assessment but also facilitate more precise decision-making regarding neoadjuvant and adjuvant therapies [103,104]. In parallel, understanding tumor–stromal interactions, particularly mechanisms such as VEGF-C/D–mediated lymphangiogenesis and CAF-driven microenvironmental remodeling, offers translational insights into surgical and perioperative management. For example, CAFs promote lymphatic endothelial proliferation and lymph node metastasis via paracrine PDGF-BB/PDGFR-β signaling, and such activity correlates with more extensive nodal spread potentially necessitating broader lymphadenectomy or adjunctive therapy in high-risk cases [84]. Recognizing these biological drivers can help stratify patients who may benefit from extended lymph node dissection and tailor the timing and intensity of systemic treatment, particularly in the neoadjuvant setting [84]. These findings imply that the value of lymph node dissection may be influenced by the timing and modality of systemic treatment, highlighting the need for treatment sequencing tailored to the LNM risk and tumor stage. Future research should continue to elucidate the precise crosstalk between tumor cells and the bile-influenced TME while harnessing these advanced models to develop novel targeted therapies and optimized neoadjuvant strategies tailored to GBC patients with LNM.

## 5. Conclusions

LNM remains a major challenge in the clinical management of GBC due to its strong association with poor prognosis and limited therapeutic options. This review highlights the multifactorial nature of lymphatic dissemination in GBC, involving a complex interplay between soluble factors, stromal cells, ECM components, metabolic signals, and ncRNAs within the TME. These elements collectively drive lymphangiogenesis, immune evasion, and tumor cell migration. Emerging technologies such as single-cell sequencing, spatial transcriptomics, and patient-derived organoids offer powerful tools to dissect the spatial and temporal dynamics of these processes. A deeper understanding of these molecular mechanisms is essential for identifying new biomarkers and developing targeted strategies to disrupt early metastatic spread, ultimately improving outcomes for patients with GBC.

## Figures and Tables

**Figure 1 biomedicines-13-01372-f001:**
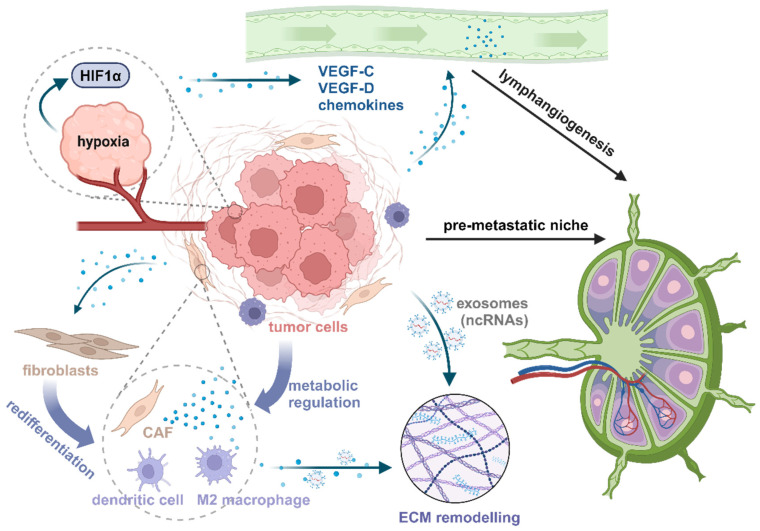
Mechanisms of lymph node metastasis in gallbladder cancer. Illustration of key molecules involved in LNM in GBC. Tumor cells promote lymphangiogenesis and the formation of a pre-metastatic niche through hypoxia-inducible factors, secretion of VEGFs, chemokines, and exosomes. CAFs, macrophages, and other tumor immune-related cells in the TME contribute to ECM remodeling and support metastatic spread. CAF, cancer-associated fibroblast; ECM, extracellular matrix; Created in BioRender. Tang, Q. (2025) https://BioRender.com/r86o924 (accessed on 28 April 2025).

**Figure 2 biomedicines-13-01372-f002:**
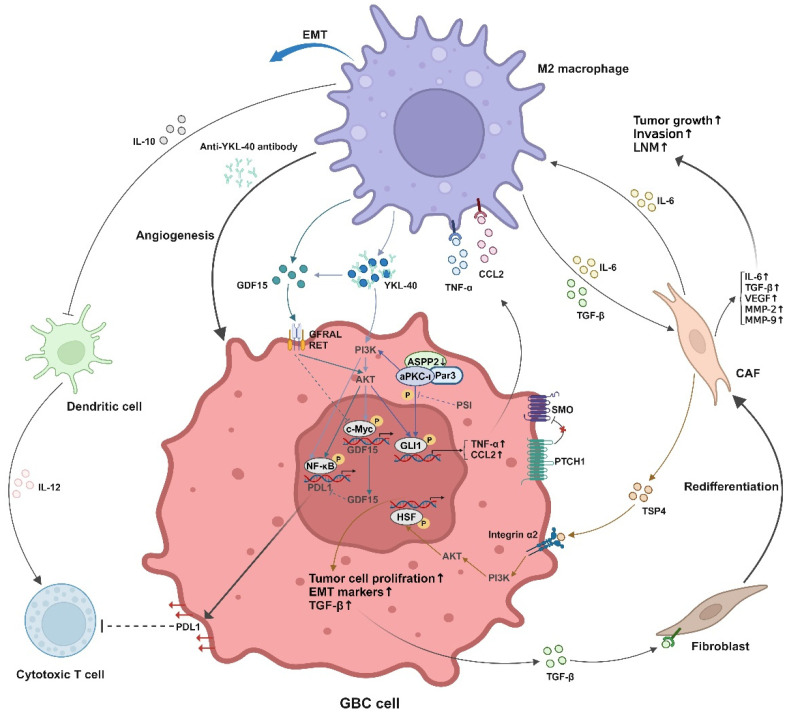
Interplay between tumor microenvironment components and signaling pathways driving oncogenic progression in gallbladder cancer. M2 macrophages promote tumor growth, invasion, and LN metastasis via IL-10 secretion and anti-YKL-40 antibody-mediated modulation, while upregulating IL-6, TGF-β, and MMPs to enhance extracellular matrix remodeling and EMT. CAFs contribute to tumor cell proliferation and EMT marker elevation through TGF-β and CCL2 signaling. The PI3K/AKT axis and integrin α2 further potentiate metastatic behavior. GDF15-GFRAL interactions may regulate metabolic reprogramming, and elevated PD-L1 expression on tumor cells suggests immune evasion by suppressing cytotoxic T cell activity. Dendritic cells and SMO/PTCH1 pathways highlight additional crosstalk within the stromal compartment. AKT, protein kinase B; CAF, cancer-associated fibroblast; CCL2, C-C motif chemokine ligand 2; EMT, epithelial–mesenchymal transition; GBC, gallbladder cancer; GDF15, growth differentiation factor 15; GFRAL, IL-6, interleukin-6; IL-10, interleukin-10; IL-12, interleukin-12; LNM, lymph node metastasis; MMP, matrix metalloproteinase; PD-L1, programmed death-ligand 1; PI3K, phosphoinositide 3-kinase; PTCH1, patched 1; SMO, smoothened; TGF-β, transforming growth factor-beta; TNF-α, tumor necrosis factor-alpha; YKL-40, chitinase 3-like protein 1. Created in BioRender. Tang, Q. (2025) https://BioRender.com/yuva1qr (accessed on 28 April 2025).

**Table 1 biomedicines-13-01372-t001:** Key non-coding RNAs in gallbladder cancer progression and lymph node metastasis.

ncRNA Type	Name	Expression	Correlation with LNM	Key Mechanisms	Refs.
lncRNA	H19	Upregulated	Positive	Promotes EMT via miR-342-3p/Wnt/β-catenin axis	[60,61]
	LINC00662	Upregulated	Positive	Drives EMT through LINC00662/miR-335-5p/OCT4 axis	[62]
	GCASPC	Downregulated	Negative	Suppresses tumor growth and LNM	[63]
	MEG3	Downregulated	Negative	Inhibits EMT and invasiveness	[63]
miRNA	miR-324-5p	Downregulated	Negative	Inhibits migration, invasion, and EMT by targeting TGF-β2	[64]
	miR-141	Upregulated	Positive	Promotes proliferation and suppresses apoptosis	[65]
circRNA	circPVT1	Upregulated	Positive	Sponges miR-339-3p to upregulate MCL1, enhancing invasiveness	[66]
	circTP63	Upregulated	Positive	Induces EMT via miR-217 sequestration	[67]
	circAATF	Upregulated	Positive	Elevates PD-L1 via miR-142-5p sponging, promoting immune evasion	[68]
	circEZH2	Upregulated	Positive	Upregulates SCD1 via miR-556-5p sponging, reprogramming lipid metabolism	[69]
	circNGFR	Upregulated	Positive	Suppresses ferroptosis to sustain tumor survival	[70]

Abbreviations: EMT, epithelial–mesenchymal transition; LNM, lymph node metastasis; ncRNA, non-coding RNA; PD-L1, programmed cell death-ligand 1; SCD1, stearoyl-CoA desaturase-1; TGF-β2, transforming growth factor beta 2.

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
