# Peer review of "Molecular Mechanisms of Lymph Node Metastasis in Gallbladder Cancer: Insights into the Tumor Microenvironment"

_biomedicines, 2025, doi:10.3390/biomedicines13061372_

Round 1

Reviewer 1 Report

Comments and Suggestions for Authors

Lymph node metastasis in gallbladder cancer (GBC) involves a complex interplay of molecular mechanisms that facilitate the spread of cancer cells from the primary tumor to regional lymph nodes. The process of lymph node metastasis in gallbladder cancer is multifactorial, involving changes in cellular behavior, interactions with the microenvironment, and immune evasion strategies. Understanding these mechanisms is crucial for developing targeted therapies aimed at preventing or treating metastatic disease. Further research into these pathways may lead to novel therapeutic strategies to improve outcomes for patients with gallbladder cancer. Therefore, the paper of Dr.Tang et al is very important.

Comments

  1. All abbreviations should be disclosed at first use. The role of every described growth factor, chemokine etc. should be described in detail for clarity and its importance in the processes described. This should be corrected.
  2. The role of Epithelial-Mesenchymal Transition (EMT), Regulatory T Cells (Tregs), Circulating Tumor Cells (CTCs), • Mutations in oncogenes (e.g., KRAS, TP53) and tumor suppressor genes was not addressed. This should be corrected.
  3. Section 1.3: The role of hypoxia is not clear. The authors should present their hypothesis of how it can work in the tumour environment.
  4. Section MMPs should be corrected as something is missing there. In addition , the authors should present their hypothesis of how it can work in the tumour environment.
  5. Section Integrins: The data here contradicts to that in the section MMPs. This should be corrected.
  6. Section fibronectins: The mechanism, how fibronectins can affect cell proliferation is not clear. Is there any role of fibronectin fragments? This should be clarified.
  7. Exosomes section : Here MMPs and integrins are definitely involved in cancer process while in the MMPs section this involvement is contradictory. The authors should remove all the contradictory statements.
  8. Circular RNAs section: The authors should describe in detail the putative mechanisms of ferroptosis and lipid metabolism involvement in GBC.
  9. In the last sentence of Circular RNAs section it should be indicated which author belief is presented and whether the co-authors share this belief.
  10. Section 1.7 should be described in more detail.
  11. Section 2.2: It is not clear, how the authors suggest to use M2 downregulation as a potential therapy as it will result in M1 macrophage upregulation and consequent increase in inflammation. This should be clarified.
  12. Discussion: The first paragraph should be moved to Introduction section.
  13. Conclusion: This section should not contain any references. The data which the authors are referred for should be moved to the Discussion section/.
  14. In general, the review gives the impression of being superficial and not thought out.

Author Response

Comments 1: All abbreviations should be disclosed at first use. The role of every described growth factor, chemokine etc. should be described in detail for clarity and its importance in the processes described.

Response 1: We are sorry to make abbreviation mistakes and sincerely thank the reviewer for this helpful comment. Following your suggestion, we have carefully reviewed the entire manuscript to ensure that all abbreviations are defined at first use (Page 4, line 103; Page 5, line 182; Page 6, line 188, 191; Page 11, line 431-432). Beyond this formatting clarification, we have taken the opportunity to strengthen mechanistic clarity in several key sections:

VEGF-C/D section (Page 4, line 115-120): We expanded our discussion of how VEGF-D promoted lymphangiogenesis and highlighted its role in facilitating GBC cell invasion and lymphatic metastasis. Based on the findings from Lin et al., we further added in vivo evidence that VEGF-D silencing in NOZ xenograft models significantly reduces tumor growth, ascites formation, hepatic invasion, and lymphatic spread, reinforcing its role as both a lymphangiogenic and metastasis-promoting molecule. This addition helps clarify why VEGF-D is not only a marker but a therapeutic target candidate in GBC.

Chemokine section (Page 4, line 136-138): Descriptions of pathways including CCL2/CCR2 and CXCL5/CXCR2 were rewritten to explain not only their expression patterns but also their downstream signaling relevance to lymph node metastasis.

Hypoxia section (Page 5, line 174-181): We expanded the mechanistic explanation of how hypoxia-inducible factor 1-alpha (HIF-1α) enhances VEGF-C transcription and drives lymphangiogenesis and immune escape in the TME of GBC.

Cell Adhesion and Migration Signaling section (Page 6, line 199-203, 227-232): We added mechanistic details on how matrix metalloproteinases and fibronectins contribute to lymphatic dissemination.

We believe these changes greatly enhance the biological clarity and functional coherence of the review, and we thank the reviewer for prompting us to strengthen these important areas.

Comments 2: The role of Epithelial-Mesenchymal Transition (EMT), Regulatory T Cells (Tregs), Circulating Tumor Cells (CTCs), mutations in oncogenes (e.g., KRAS, TP53) and tumor suppressor genes was not addressed.

Response 2: Thank you for this insightful suggestion. We have thoroughly reviewed the literature and found that while EMT is a common downstream process among several mechanisms discussed in our review (e.g., CAFs, chemokines, ncRNAs), direct studies on the roles of Tregs, CTCs, or mutations in KRAS and TP53 in the context of GBC LNM remain scarce. To acknowledge this research gap while respecting the reviewer’s comment, we have added statements to the Discussion (Page 12, line 454-458) highlighting EMT as a cross-cutting mechanism and noting the lack of specific evidence on Tregs, CTCs, and oncogenic mutations in GBC (Page 12, line 466-471). We now explicitly state that these elements are well characterized in other cancers but require further validation in GBC. We hope this clarification aligns with the reviewer’s expectations.

Comment 3: Section 1.3: The role of hypoxia is not clear. The authors should present their hypothesis of how it can work in the tumour environment.

Response 3: We sincerely appreciate your suggestion, which helped improve the clarity and focus of this section. In the revised Section 1.3 (Page 5, line 161-181), we have added a working hypothesis suggesting that hypoxia may contribute to LNM in GBC via two possible mechanisms: (1) promoting EMT and (2) activating the HIF-1α/VEGF axis to enhance lymphangiogenesis. We also clarified the contrasting findings in current literature and emphasized the need for further research to validate these pathways. We hope this addresses your concern.

Comment 4: Section MMPs should be corrected as something is missing there. In addition, the authors should present their hypothesis of how it can work in the tumour environment.

Response 4: Thank you for pointing this out. We agree that the original section lacked a clear mechanistic hypothesis. In the revised manuscript (Page 6, line 199-204), we have added a detailed discussion of how MMPs may contribute to LNM in the GBC tumor microenvironment. Specifically, we hypothesize that MMPs facilitate lymphatic invasion by degrading ECM barriers and enabling tumor cell migration toward lymphatic vessels, potentially regulated by VEGF-C and inflammatory stimuli. We hope this addresses the reviewer’s concern and provides a more comprehensive understanding of the role of MMPs in GBC metastasis.

Comment 5: Section Integrins: The data here contradicts that in the section MMPs. This should be corrected.

Response 5: We thank the reviewer for this important concern. We would like to clarify that these sections describe findings from different relevant cancer types, since direct mechanistic evidence linking MMPs to LNM in gallbladder cancer is limited, although supportive data exist in related cancers like intrahepatic cholangiocarcinoma. In the Integrins section, the role of integrin β6 in promoting MMP-9 expression and LNM is drawn from studies in cholangiocarcinoma. We have revised the Integrins section (Page 6, line 213-215) to explicitly note that while the integrin–MMP axis is well-established in cholangiocarcinoma, its relevance in gallbladder cancer remains lack of research. We hope we clarified the perceived contradiction.

Comment 6: Section fibronectins: The mechanism, how fibronectins can affect cell proliferation is not clear. Is there any role of fibronectin fragments? This should be clarified.

Response 6: We thank the reviewer for pointing out the need to clarify the mechanism by which FN affects tumor cell proliferation and the potential role of FN fragments. We clarified the role of fibronectin in activating Akt/mTOR/4E-BP1 signaling to promote proliferation. We also incorporated new findings indicating that proteolytic FN fragments containing RGD and synergy motifs enhance integrin binding and downstream signaling. These updates (Page 6-7, line 227-234) provide mechanistic clarity and highlight the functional role of FN fragments in the tumor microenvironment.

Comment 7: Exosomes section: Here MMPs and integrins are definitely involved in cancer process while in the MMPs section this involvement is contradictory. The authors should remove all the contradictory statements.

Response 7: We thank the reviewer for the comment and appreciate the opportunity to clarify this point. In the Exosomes section, our discussion of MMPs and integrins is explicitly based on evidence from other cancer types, such as bladder cancer, and that similar mechanisms are hypothesized but not yet validated in gallbladder cancer, since direct mechanistic evidence linking exosomes to LNM in gallbladder cancer is limited. We acknowledge that our original words may have given the impression that these mechanisms were established in gallbladder cancer, which was not our intention. To avoid any misunderstanding, we have revised the corresponding paragraph (Page 7, line 249-251) to clearly indicate that these mechanisms are drawn from non-gallbladder cancer contexts and serve as hypotheses rather than confirmed pathways in gallbladder cancer. We have also added a statement explicitly noting that while these findings are compelling, their relevance to gallbladder cancer remains to be validated. We believe this clarification resolves the perceived contradiction.

Comment 8: Circular RNAs section: The authors should describe in detail the putative mechanisms of ferroptosis and lipid metabolism involvement in GBC.

Response 8: Thank you for the insightful comment. We have revised the paragraph (Page 9, line 310-318) to provide a more detailed and mechanistic explanation of how circEZH2 and circNGFR regulate lipid metabolism and ferroptosis in GBC. The updated version includes the roles of miR-556-5p/SCD1, IGF2BP2-mediated stabilization, and the HO-1/FTH axis, offering a clearer link between circRNA activity and ferroptosis modulation. We hope this expanded description better addresses the reviewer’s concerns.

Comments 9: In the last sentence of Circular RNAs section it should be indicated which author belief is presented and whether the co-authors share this belief.

Response 9: We revised the last sentence (Page 9, line 328) to clarify that this is a collective conclusion by all co-authors, based on literature and synthesis of current data.

Comments 10: Section 1.7 should be described in more detail.

Response 10: Thank you for your valuable comment. We agree that Section 1.7 lacked sufficient mechanistic detail and made more detailed explanations. Given the limited high-impact publications directly focused on GBC, we have also now expanded the section by incorporating mechanistically relevant research on lipid metabolism in cholangiocarcinoma and hepatocellular carcinoma, which share similar metabolic and anatomical contexts (Page 10, line 354-355, 358-367). These additions provide deeper insights into how metabolic remodeling promotes lymph node metastasis through immune suppression, fatty acid oxidation, and oncogenic signaling.

Comments 11: Section 2.2: It is not clear, how the authors suggest to use M2 downregulation as a potential therapy as it will result in M1 macrophage upregulation and consequent increase in inflammation. This should be clarified.

Response 11: We sincerely thank the reviewer for highlighting this important point. We fully agree that a simplistic approach to suppressing M2 macrophages may inadvertently shift the macrophage population toward a pro-inflammatory M1 phenotype, which could exacerbate systemic inflammation and tissue damage. To address this, we have revised the last paragraph of Section 2.2 (Page 11, line 426-429) to clarify that our therapeutic suggestion is not to downregulate M2 macrophages, but rather to reprogram their tumor-promoting functions. In particular, we now emphasize that targeting specific immunosuppressive factors secreted by M2 macrophages—such as YKL-40 or CCL2—or interfering with the metabolic pathways that sustain M2 polarization, may offer a more refined and safer approach to inhibiting LNM in GBC. This would allow the preservation of macrophage immune surveillance functions while reducing their pro-metastatic influence in the TME. We hope this clarification adequately addresses the reviewer’s concern.

Comments 12: Discussion: The first paragraph should be moved to Introduction section.

Response 12: We sincerely thank the reviewer for this important point. The initial paragraph of the Discussion section has been revised and moved to the appropriate place (Page 1-2, line 38-47) in the Introduction section to provide a clearer background and structure.

Comments 13: Conclusion: This section should not contain any references. The data which the authors are referred for should be moved to the Discussion section.

Response 13: We sincerely thank the reviewer for this important comment. We removed all references from the Conclusion and shifted those contents into the Discussion. The Conclusion now only summarizes key findings (Page 13, line 503-513).

Comment 14: In general, the review gives the impression of being superficial and not thought out.

Response 14: We are sorry to hear that the review was perceived as superficial, and we thank the reviewer for the opportunity to improve the manuscript. We have carefully revisited and revised multiple sections to enhance clarity, mechanistic depth, and structural coherence. We appreciate the candid feedback and have extensively revised multiple sections, including Hypoxia, EMT, Immune Cells, ECM remodeling, and ncRNA subsections. We improved the depth of mechanistic insights, added missing elements, clarified contradictions, and restructured the manuscript to present a more coherent, thoughtful review. We hope these changes have significantly improved the scientific value and clarity of the paper.

Importantly, we would like to express that one of the aims of this review is to provide a focused and disease-specific summary of mechanisms underlying LNM in GBC. We believe that identifying both well-supported pathways and areas of research gap (such as the roles of Tregs and CTCs) is crucial for advancing this field. This emphasis on mapping current knowledge boundaries and highlighting unmet questions is one of the key strengths and contributions of our manuscript. We hope the revised version more clearly reflects the value of this focused and forward-looking approach.

Reviewer 2 Report

Comments and Suggestions for Authors

Congratulations to Tang et on on this article. This is a well-written and insightful review that explores the complex molecular mechanisms underlying lymph node metastasis (LNM) in gallbladder cancer (GBC), a topic of significant clinical importance given the aggressive nature of this malignancy and its poor prognosis when nodal involvement is present. The manuscript is comprehensive, clearly structured, and supported by high-quality, didactic figures that enhance the reader's understanding of the tumor microenvironment’s role in facilitating LNM, including the contributions of VEGFs, chemokines, CAFs, TAMs, HIFs, and ncRNAs.

The authors also highlight potential therapeutic targets and future directions, highly relevant for advancing precision medicine.

I believe the manuscript merits publication following minor revisions.

I suggest the authors expand the clinical implications of their findings by discussing how a better understanding of lymphatic dissemination mechanisms in GBC could influence surgical strategies, particularly regarding the extent, indication, and anatomical rationale for lymphadenectomy. Including a description of the primary lymph node stations involved in GBC spread would provide valuable context for surgical decision-making and enhance the translational relevance of the review.

Lastly, a minor point: there is an extraneous “v” at the end of the abstract that should be removed.

Author Response

Comment 1: I suggest the authors expand the clinical implications of their findings by discussing how a better understanding of lymphatic dissemination mechanisms in GBC could influence surgical strategies, particularly regarding the extent, indication, and anatomical rationale for lymphadenectomy. Including a description of the primary lymph node stations involved in GBC spread would provide valuable context for surgical decision-making and enhance the translational relevance of the review.

Response 1: We thank the reviewer for this insightful suggestion. We fully agree that connecting molecular insights with surgical strategies is an important aspect of translational relevance. In the revised manuscript, we have added a paragraph to the Introduction (Page 1-2, line 38-47) briefly summarizing the anatomical routes of lymphatic drainage in GBC and the primary lymph node stations involved (e.g., cystic, pericholedochal, periportal, and para-aortic nodes). We also reference a recent, comprehensive clinical review focused on surgical lymphadenectomy strategies (Li Y et al., Front Oncol, 2022), which provides an excellent summary of surgical lymphadenectomy strategies in GBC. Given that our review is focused on the molecular and microenvironmental mechanisms underlying lymphatic dissemination, we believe that detailed surgical and anatomical discussion is beyond the main scope of this article. However, to strengthen the translational bridge, we have expanded the Discussion section (Page 12-13, line 487-495) to highlight how understanding mechanisms such as VEGF-C/D–mediated lymphangiogenesis or CAF-induced remodeling may influence the future refinement of lymphadenectomy scope and guide perioperative therapeutic planning. We hope this integrated approach appropriately balances clinical relevance with mechanistic depth.

Comment 2: Lastly, a minor point: there is an extraneous “v” at the end of the abstract that should be removed.

Response 2: We thank the reviewer for catching this typographical error. The extraneous “v” at the end of the abstract has been removed in the revised version.

Reviewer 3 Report

Comments and Suggestions for Authors

I appreciate the opportunity to review this interesting article. It is a review article on lymph node metastases in gallbladder cancer, with a special emphasis on the tumor microenvironment. Overall, I find the article to be well-written and adequately referenced. The authors use current references to provide an overview of what is known and unknown about the pathophysiology of metastasis and the tumor microenvironment. The detailed and fluid introduction provides a good theoretical framework that highlights the topic's relevance. The topic is developed clearly and in detail, and tables and figures are used effectively (the latter being especially useful). The discussion is concise and highlights opportunities for future research. My only comment concerns the conclusion. It should be shorter and more concise, highlighting the authors' opinion. In its current form, some references are still discussed in the conclusion, which could be moved to the discussion section. Specifically, I am referring to the comments about references 91 and 93 in the conclusion. Overall, I found the article to be excellent and interesting.

Author Response

Comment 1: My only comment concerns the conclusion. It should be shorter and more concise, highlighting the authors' opinion. In its current form, some references are still discussed in the conclusion, which could be moved to the discussion section. Specifically, I am referring to the comments about references 91 and 93 in the conclusion.

Response 1: Thank you for this helpful suggestion. In the revised manuscript, we have substantially shortened the Conclusion section to make it more concise and opinion-focused. Specifically, we have removed all references from the Conclusion (Page 13, line 503-513). The revised Conclusion now summarizes the main findings and perspectives without discussing specific studies, in line with your recommendation.

Round 2

Reviewer 1 Report

Comments and Suggestions for Authors

I have no more comments. Accept as is.